# Lung Fibrosis and Fibrosis in the Lungs: Is It All about Myofibroblasts?

**DOI:** 10.3390/biomedicines10061423

**Published:** 2022-06-15

**Authors:** Elena Ortiz-Zapater, Jaime Signes-Costa, Paula Montero, Inés Roger

**Affiliations:** 1Department of Biochemistry and Molecular Biology, Faculty of Medicine-IIS INCLIVA, University of Valencia, 46010 Valencia, Spain; 2Pulmonary Department, Hospital Clinico, INCLIVA, 46010 Valencia, Spain; jaimesignescosta@gmail.com; 3Department of Pharmacology, Faculty of Medicine, University of Valencia, 46010 Valencia, Spain; paulamonmag@gmail.com (P.M.); irola3@gmail.com (I.R.); 4Biomedical Research Networking Centre on Respiratory Diseases (CIBERES), Health Institute Carlos III, 28029 Madrid, Spain

**Keywords:** pulmonary fibrosis, myofibroblast, macrophages, secretome, EMT, endoMT, TGF-β

## Abstract

In the lungs, fibrosis is a growing clinical problem that results in shortness of breath and can end up in respiratory failure. Even though the main fibrotic disease affecting the lung is idiopathic pulmonary fibrosis (IPF), which affects the interstitial space, there are many fibrotic events that have high and dangerous consequences for the lungs. Asthma, chronic obstructive pulmonary disease (COPD), excessive allergies, clearance of infection or COVID-19, all are frequent diseases that show lung fibrosis. In this review, we describe the different kinds of fibrosis and analyse the main types of cells involved—myofibroblasts and other cells, like macrophages—and review the main fibrotic mechanisms. Finally, we analyse present treatments for fibrosis in the lungs and highlight potential targets for anti-fibrotic therapies.

## 1. What Is Lung Fibrosis?

The adult lung is one of the few organs that comes in direct contact with the outside environment. In that sense, the single layer of respiratory epithelial cells that lines the airways is constantly exposed to potentially toxic agents and pathogens and therefore is in constant communication with the mucosal immune system. For that reason, it must be able to respond quickly and effectively to repair cellular damage. The interaction between the epithelium and the other cell types in the lungs is thus crucial, and when dysregulated, produces pathological processes in a wide range of respiratory conditions, including fibrosis. 

Lung fibrosis can be defined as the aberrant activation and differentiation of lung fibroblasts to myofibroblasts, persistent alveolar injury, exuberant extracellular matrix (ECM) deposition, and incomplete repair that over time can lead to progressive loss of respiratory function and death. According to the World Health Organization (WHO), the top-three causes of death worldwide in descending order are associated with cardiovascular, respiratory, and neonatal conditions (https://www.who.int/news-room/fact-sheets/detail/the-top-10-causes-of-death (accessed on 9 December 2020)). Lower respiratory diseases rank fourth. As defined by the U.S. Centers for Disease Control and Prevention (CDC) and the WHO, chronic lower respiratory disease (CLRD) encompasses four major diseases: chronic obstructive pulmonary disease (COPD), chronic bronchitis, emphysema, and asthma [1,2]. Three of these are involved in the appearance of fibrosis in the lungs although the cell subsets and the mechanisms involved in the development of fibrotic events may differ. In that sense, asthma, COPD, and idiopathic pulmonary fibrosis (IPF) are categorised as severe multifactorial lung diseases that have the common feature of lung remodelling in morphologically distinct compartments, notably the large or small airways and parenchyma [3]. Likewise, innate and adaptive immune cells are fundamental for protection. Understanding the complex composition of cells—not only differentiated myofibroblasts but also fibroblasts, endothelial and epithelial cells, and innate and adaptive immune cells—and extracellular matrix (ECM) components such as collagen and fibronectin is important for understanding the different fibrotic events that can occur. Among the different cells, we emphasize the role of macrophages, as they have been shown to activate myofibroblasts, the main source of the extracellular matrix in fibrotic diseases.

## 2. Myofibroblasts in the Lung

The respiratory tract is a complex organ, that can be differentiated into upper and lower airways. The upper respiratory tract includes the nose and nasal passages, pharynx, and larynx. The lower respiratory tract comprises the trachea, bronchi, and bronchioles, the conducting airways where respiration occurs. They include respiratory bronchioles, alveolar ducts, alveolar sacs, and alveoli. The respiratory system is not only formed by epithelial cells but is also lined with cells from the circulatory system, resident lung cells (including immune cells) and mesenchymal cells. These are all connected thanks to the ECM.

In this scenario, myofibroblasts play an important role during tissue repair. They show an increased contractibility, hence their role in closing open wounds. However, they are the main source of ECM components, and when dysregulated, they secrete excessive amounts of ECM proteins, which alters the normal structure of the respiratory airways, eventually disrupting normal lung physiology [4].

Myofibroblasts were first described in ultrastructural examinations of rat wounds [5]. In the first description, Gabbiani et al. characterized the presence of modified fibroblasts in the granulation tissue of wounds [6]. They are large cells with ruffled membranes, and they exhibit cytoplasmic stress fibres, such as those found in contractile smooth muscle cells, along with peripheral focal adhesions [4].

The origin of these myofibroblasts is still a matter of debate, meaning that its identification would be a step toward understanding of the mechanisms responsible for fibrosis, which may lead to the discovery of new therapeutic targets for different fibrotic processes in the lung.

The contribution of resident mesenchymal populations has been broadly accepted [7,8]; moreover, immune cells, especially resident macrophages, play a major role in the activation and differentiation of myofibroblasts, which is extensively described in this review. Moreover, signals produced in response to inflammation or mechanical stresses can also trigger processes involved in the epithelial–mesenchymal transition (EMT) and endothelial–mesenchymal transition (EndoMT). Finally, we cannot forget the contribution from other cellular populations, like pericytes or fibrocytes, or the inherent biochemical and physical properties of the ECM componentsFinally, although the activation or differentiation into myofibroblasts may represent the final stage in remodelling, some authors have raised the possibility that myofibroblasts may represent a functional population.

The main cell types that both differentiate into, or activate, myofibroblasts in the lung are described in the next sections, which focus not only on their involvement in myofibroblast activation but also on the importance of these cells per se in different fibrotic processes that in lung disease.

### 2.1. Epithelial Cells

Single-cell RNA sequencing (scRNA-seq) has changed the information we have about the layer of different cells that covers the surface of the airways [9,10]. Although not the focus of this review, we summarize here and in Figure 1 the main cells found in the respiratory epithelium. Although some authors doubt that epithelial cells serve as progenitors of myofibroblasts, we believe there is enough evidence to claim that the epithelial-to-mesenchymal transition (EMT) explains myofibroblast differentiation. Moreover, there is considerable evidence showing that cells of epithelial origin contribute to fibrotic lesions and other respiratory diseases. Four main cells—basal, club, goblet and ciliated—line the conduction airways.

**Basal cells** are the progenitors of the airway epithelium. They can self-renew after injury and differentiate into other important cell types: club, goblet, and ciliated; pulmonary ionocytes; pulmonary neuroendocrine cells (PNECs); and tuft cells [11,12,13]. They have a cuboidal shape and they are attached to the basement membrane. In the human lung, they express cytokeratin 5 and 14 as well as many other markers [12,13]. Basal cell expansion is a common feature of epithelial remodelling in IPF and other chronic lung diseases.

**Club cells** were previously known as Clara cells, a term that the scientific community should have avoided as it reflects their discovery by a Nazi military doctor. These cells have a dome shape and are the main cell type in respiratory bronchioles [14]. Club cells are commonly characterized by the expression of Scgb1a1 [15], and like basal cells, also behave as stem cells and aid in epithelial repair [16]. The dysregulation of club cells contributes to many respiratory conditions, including asthma, IPF, cystic fibrosis, pulmonary fibrosis or COPD [17,18].

**Goblet cells** are present in airway lining and secrete a gel-forming mucus [19]. Together with ciliated cells, they facilitate effective mucus clearance by expressing mucins, mainly MUC5AC and MUC5B. Increased mucus production by Goblet cells is a typical feature of many respiratory diseases, including asthma, COPD, primary cilia dyskinesia (PCD) or cystic fibrosis (CF) [9,20].

**Ciliated cells** are columnar cells that are found throughout the airways. They play a pivotal role in airway homeostasis and together with the airway mucus mediate the critical mucociliary clearance (MCC) of inhaled particles and pathogens [21]. Chronic lung disorders are characterised by impaired MCC. For example, it has been shown that smoking shortens the size of the cilia. COPD patients also show a significant decrease in the number of ciliated cells [22,23,24].

Basal, club, goblet and ciliated cells are the main ones in the lining of the conducting airways and the most involved in the EMT, which produces myofibroblasts. Apart from these, other epithelial cells are also present in the airways but in a very low proportion. However, they include PNECs, which behave as stem cells [22] and have been involved in inducing the proliferation and migration of fibroblasts [25] or other minority cells, such as tuft cells, ionocytes or Hillock cells (see Figure 1).

The main known function of PNECs is to serve as a key element in the crosstalk between the immune and the nervous systems [14]. They sense airway activity and secrete neuropeptides (serotonin, calcitonin gene-related peptide) to stimulate immune responses [26,27]. **Tuft cells** have been described in both the gastrointestinal tract [28] nose, trachea, and proximal airways, but their role in the lungs is less clear, so evidence implicating them in respiratory disease is limited [29,30]; however, it has been suggested that they decrease the effects produced by an airway allergic reaction. Rane et al. also demonstrated that they develop in the lung after a severe influenza injury [31]. **Pulmonary ionocytes** express CF transmembrane conductance regulator (*CFTR*) and forkhead box I1 (*FOXI1*) [32,33], and because of that there is a hope that they could be used to develop a CF treatment [29]. Finally, there is not much information about **hillock cells**; in fact, it is not apparent that they are present along the whole respiratory tract. They were first described by Montoro et al., as a different kind of club cell that express keratin 13 [32]. They are normally found in groups that form structures that resemble small hills, hence their name. Their function is not clear, but it is known that they have a particularly high turnover rate [29] and express high levels of Ki67 [9]. 

**Alveolar type I and II cells:** Pulmonary alveolar type I (AT1) cells cover the great majority of the alveolar surface. They are essential for alveolar regeneration and are in charge of the air–blood barrier function. However, it is definitively the AT2 population that is more involved in fibrosis, especially IPF. In that sense, there are numerous studies showing that AT2 injury or malfunction triggers IPF. As a consequence, fibroblast proliferation increases as it does paracrine signalling to myofibroblasts, the production of other pro-fibrotic factors, and the development of fibrotic processes [25], leading to the progressive loss of lung function [26,27]. Moreover, constant damage to these cells produces impaired repair function. In that sense, AT2 cells in fibrotic lesions soon become hypertrophic and hyperplastic [28].

### 2.2. Endothelial Cells

The main process by which endothelial cells differentiate into myofibroblasts is via endothelial–mesenchymal transition (EndoMT). This process shares many characteristics with the EMT, and in fact has been described as a different kind of EMT. The EndoMT is characterised by the loss of typical endothelial markers such as CD31/PECAM-1 (platelet EC adhesion molecule-1), Tie1, Tie2, VE-cadherin and von Willebrand Factor (vWF); and the expression of mesenchymal markers such as such as α-SMA, N-cadherin and vimentin [29]. Recently, Kovacic et al. [30] reviewed the signalling pathways controlling EndoMT and found that transforming growth factor beta (TGF-β) was one of the main ones. 

Apart from this, endothelial cells also contribute to fibrosis by producing profibrotic molecules as well as chemokines and cytokines that recruit immune cells that contribute to inflammation of the fibrotic foci. In the lung, pulmonary capillary endothelial cells, through EndoMT, may serve as a source of fibroblasts in pulmonary fibrosis [31].

### 2.3. Resident Mesenchymal Cells: Fibroblasts

Fibroblasts play a major role in the process of fibrosis in the lungs as they are responsible for the production and homeostasis of the ECM. Therefore, they modify the composition and the properties of the ECM and, very importantly, differentiate into myofibroblasts, one of the main cell types associated with the development of fibrotic tissue [32]. Fibroblasts are regulated by different molecules: they can be influenced by ECM components and other molecules present in the matrix, such as receptors, cytokines, chemokines, and growth factors. Among these, TGF-β, is the main cytokine that stimulates fibroblasts to produce ECM constituents (see part 4). However, molecules such as interferon-gamma (IFN-γ) or tumour necrosis factor (TNF) [33,34,35,36,37,38] also act as fibroblast inhibitors. As shown in Figure 2 and part 4, fibroblasts are one of the major sources of myofibroblasts under pathological conditions when ECM turnover is altered and fibrosis is found in the bronchi and bronchioles [33,39,40,41,42].

### 2.4. Bone Marrow/Derived Precursors: Fibrocytes

Fibrocytes have also been observed in human fibrotic lung disease. They are spindle-shaped mesenchymal progenitor cells that were first described by Bucala et al. [43]. They express markers such as Col-1 and CD45 [44] and have been described in fibrotic disorders not only in the lung but also in other organs. These cells respond to signals and transmigrate to damaged sites where they differentiate into myofibroblasts [45]. It has been proposed that this happens through factors such as interleukin-31 or stem cell factor (SCF) [46]. In vitro, TGF-β stimulated fibrocytes in monolayers that then differentiate into cells expressing α-SMA and causing contraction of collagen gels [47].

Apart from this role in differentiation, they also produce ECM components, such as collagen I and III, vimentin and fibronectin [48]. Another mechanism by which fibrocytes may indirectly influence fibroblastic behaviour in resident cells is by the release of exosomes [4]. Finally, fibrocytes are thought to activate fibroblasts or inherently participate in inflammation, which of course, leads to ECM remodelling and fibrosis [49,50].

Fibrocytes have been described in patients with IPF, specifically in the fibrotic lesions [51]. Moreover, there is an increase of the concentration of fibrocytes in IPF patients, which has been shown to be a sign of poor prognosis [52].

It is important to note that, as Byrne et al. stated, it is not very clear whether fibrocytes represent a distinct, fibroblast-precursor lineage or are (as is more likely) a subset of monocyte-derived macrophages, which again underlining the extent of macrophage plasticity [53].

### 2.5. Perivascular Stromal Origin: Pericytes

Lung pericytes are mesenchymal-derived cells in the basement membrane of capillaries that have been shown to add structural support to small vessels, as well as control vascular tone. They also produce molecules that promote angiogenesis and components of the ECM [54]. In fact, the loss of pericytes in the alveolar capillaries has been associated with lung fibrosis. Lung pericytes are also capable of responding to danger signals and amplifying the inflammatory response through the elaboration of cytokines and adhesion molecules [4].

The first paper describing pericyte involvement in myofibroblasts differentiation was by Göritz et al. [55], in which they described pericyte function in wound healing in the central nervous system. In the lung, Hung et al. [56] used the bleomycin model to demonstrate that pericytes are present in fibrotic lesions.

## 3. Role of Macrophages in Myofibroblasts Differentiation and Activation

If there is one immune population that is important to highlight in the different processes of fibrosis in the lungs, it is macrophages. To start with, they are versatile cells that exhibit a high degree of plasticity. When the lung is injured, macrophages are pushed towards a pro-inflammatory phenotype that promotes differentiation and activation of myofibroblasts. Macrophages are the main innate immune cells in the healthy lung and have many functions: immune surveillance, cellular debris removal, microbe clearance, inflammation resolution, and repair [57]. In addition, they localize close to myofibroblasts, which are considered the major ECM source during fibrosis [58]. Moreover, an elevated expression of macrophage-derived molecules has been found in fibrotic lungs [59].

As will be described in Table 1, macrophages secrete many different molecules important for the development of fibrosis. Moreover, as we have learned so far, myofibroblasts are thought to come from many different progenitor cells. Macrophage–myofibroblast transition (MMT) is a recent term that describes the mechanism by which macrophages derived from bone marrow-derived monocytes transform into myofibroblasts, contributing to fibrosis [60]. This has been described mainly in kidney fibrosis [60,61] but also very recently in the lung [62]. The molecular mechanism that controls the MMT is still unknown, but the idea of this differentiation opens a new avenue that could bring valid targets and provide insight into the development of novel antifibrotic therapies.

There are two major macrophage subsets: alveolar macrophage (AM), and interstitial macrophage (IM). A potential third subset found in airway lumen, airway macrophages, has also been proposed and may be discovered in airways diseases such as asthma [63] or in allergic-airway mice models [64]. In that sense, Tamò et al. suggested that lung stem cells could contribute to the appearance of a new subpopulation of macrophages after fibrosis in the lungs or a repetitive injury [65].

In a healthy lung, AMs are present in larger numbers than IMs (up to eight-fold) in digested murine lungs according to Gibbing et al. [66]. Even though both types present bona fide macrophages markers, (e.g., CD64 or F4/80, CD68, Lamp1, Lysozyme 2 or CD36) [67], they also express different markers, as was very well summarized in a recent review by Shi et al. Apart from this difference in surface markers, both kinds of macrophage are also be differentiated by key transcription factors although AM macrophages have been studied more in this regard [57,66].

AMs are in the airway space, usually close to the alveolar epithelium, facing the external environment [53]. AMs are long-lived cells and under homeostatic conditional or after cell depletion, their repopulation occurs by in situ proliferation rather than by replenishment from bone marrow [68]. However, after infection or an inflammatory insult, AMs are rapidly depleted and republish by monocyte-derived AMS [69,70,71]. The work of Misharin’s group, which performed scRNA-seq to compare eight normal human lungs to eight severely fibrotic lungs, showed that AMs show greater heterogeneity in fibrotic foci [72].

Conversely, IMs reside in the lung parenchyma [53], and their functions are less studied and therefore less well understood [57]. However, their location is ideal for influencing all fibrotic processes compared to AMs. In fact, different mice studies have highlighted the importance of IMs in pulmonary fibrosis in models of radiation-induced pulmonary fibrosis [73] and airway allergic inflammation.

Apart from this difference in AMs and IMs, lung macrophage populations have been classified according to their activation status between M1 and M2. The two M1/M2 macrophages “entities” have been very much used to classify these monocyte-derived macrophages, especially in other organs such as the intestine In the lungs, however, the macrophage population is so heterogenous that it is almost impossible to classify them according to this very strict pair [74]. This is rather controversial as even though some authors doubt about the utility of describing macrophages as M1 or M2 in the context of pulmonary fibrosis [53], others have extensively described how regulators of M2 polarization contribute to fibrotic progression [75]. For example, Cheng et al. recently wrote that promoting the apoptosis of M2 macrophages or inhibiting their polarization may be beneficial for treating pulmonary fibrosis [76].

### 3.1. Macrophage-Derived Secretory Proteins and Fibrosis in the Lungs: The Macrophage Secretome

In any case, macrophages are without doubt major drivers of fibrosis and are often found close to the myofibroblasts that produce components of the ECM. They produce different chemokines and cytokines, including matrix metalloproteases, as well as pro-fibrotic soluble mediators involved in the proliferation of fibroblasts, conversion to myofibroblasts, and recruitment of other immune cells to the injury site. We have summarised the different secretory proteins produced by macrophages, in Table 1. 

**Table 1 biomedicines-10-01423-t001:** Macrophages derived secretory proteins and fibrosis in the lungs: the macrophage secretome.

Secretome Molecules	Macrophage Type	Study Subject	Role in Fibrosis	Reference
KC	Monocyte-derived AMs	LPS stimulation	Increase proliferation of ECM components	[69,77]
type IV collagenase	AMs	BLM-induced fibrosis mice model	Abnormal collagen degradation	[78,79]
MMP-1	lung macrophages	IPF, COVID-19	Role in IPF presently unknown. Present in early fibrotic processes in COVID-19 patients	[26,80]
MMP-2	lung macrophages	IPF	Effect on fibrocytes: Tissue migration and homing	[81]
MMP-3	AMs	IPF patients, lung	Induction of epithelial-mesenchymal transition through activation of β-catenin signaling	[82]
MMP-7	lung macrophages	IPF, COVID-19	Effect on AEC2 cells. Present in early fibrotic processes in COVID-19 patients	[26,80]
MMP-8	lung macrophages	IPF patients, lung	Effect on fibrocytes: Tissue migration and homing. Initiation of collagen destruction and remodelling	[81,83,84]
MMP9	AMs IMs	BLM-induced fibrosis mice model IPF patients, lung	Degradation and remodeling of extracellular matrix components	[78,84,85]
MMP-12	macrophages	LPS stimulation	Effect on TGF-β1 signaling pathway activation	[86,87]
MMP-13	Monocyte-derived macrophages e.	Lung fibrosis	Cleaves fibrillar collagens. Specific role largely unknown.	[76,87,88]
Periostin	lung macrophages	IPF, COVID-19	Present in early fibrotic processes in COVID-19 patients	[80]
TIMP1	IMs	IPF patients, lung	Promotes the fibrotic response	[89]
MT1-MMP	macrophages	IAV sensing	ECM remodelling through collagenase activity	[90]
CCL18	AMs	Pulmonary fibrosis	Increases collagen production	[91]
IFNβ	AMs	IAV sensing	Not specified	[92]
IFNγ,	M1 macrophages macrophage	Lung injury	Macrophage polarization	[93]
TGF-β1	M2 macrophages, AM	Lung fibrosis, BLM-induced fibrosis mice model, COPD patients	Differentiation of fibroblasts into myofibroblasts. EMT transition through the TGF-Smad2 signalling pathway	[76,94,95]
TNFα	AM, M1 macrophages	COPD patients, enhanced lung injury, BLM-induced fibrosis mice model	Initiates inflammation and enhaces lung injury. Induces M1 macrophages to produce proinflammatory cytokines	[76,94,96]
PDGF	primary macrophages	inflammatory process, Lung injury, lung fibrosis	Promotes de fibrotic process	[76,97]
VEGF	primary macrophages	inflammatory process, Lung injury	Promotes blood vessel development and cellular proliferation	[76,98]
IGF-1	primary macrophages	inflammatory process, Lung injury	Promotes blood vessel development and cellular proliferation	[76]
amphiregulin	primary macrophages, AM	inflammatory process, LPS stimulation, Lung injury	Contributes to immune regulation, Protects lung tissues. Counteracts epithelial damage. Enhances EGFR signalling and proliferative responses.	[99,100,101]
CCL2	macrophages	BLM-induced fibrosis mice model, IPF patients, BALF	Cellular recruitment to the lung. Enhancement of cytokine and collagen production.	[102,103]
CCL24	macrophages	BLM-induced fibrosis mice model. IPF patients, BALF	Not specified.	[102,103]
arginase 1	Monocyte-derived macrophages	Lung fibrosis	Effect on collagen synthesis	[76,104]
IL-4	M2 macrophages	Lung fibrosis		
IL-6	AM, M1 macrophages	Viral infection, enhanced lung injury	Promotes macrophages recruitment to the lung	[76,105]
IL-8	AM	COPD patients	Promotes neutrophil recrutiment	[94]
IL-10	M2 macrophages	Lung fibrosis	Induces reprogramming of macrophages to the M2 phenotype. Enhances tissue repair and promotes matrix synthesis.	[76]
IL-12	M1 macrophages	Lung injury	Not specified	[93]
IL-1β	IMs, AMs	Uninjured lung, IPF patients, lung	Triggers the activation, proliferation and transdifferentiation of epithelia cells and resident fibroblasts into myofibroblasts. Stimulates the secretion of neutrophil-attracting CXC chemokines.	[101]
IL-18	AMs	IPF patients, lung	Not specified	[106]

Abbreviations: TNF-α: Tumour necrosis factor-α; IFN-γ: Interferon-γ; LPS: Lipopolysaccharide; IL: interleukin; PDGF: platelet-derived growth factor; VEGF: vascular endothelial growth factor; IGF-1: insulin-like growth factor 1; KC: keratinocyte chemokine; MMP: Metalloproteinase; TIMP-1: Metalloproteinase Inhibitor-1; TGF-β1: Transforming growth factor 1; IPF: Idiopathic pulmonary fibrosis; COPD: Chronic obstructive pulmonary disease; BLM: Bleomicin; BALF: Bronchoalveolar lavage fluid; AM: Alveolar macrophage; IM: Interstitial macrophage; IAV: Influenza A virus; EMT: Epithelial to mesenchymal transition; ECM: Extracellular matrix.

### 3.2. Other Immune Cells

Apart from macrophages, other immune cell populations have been implicated in fibrosis and in contributing to the fibrotic microenvironment. Neutrophils, for example, behave as friends or foes in an immune response in the lung, yet numerous articles show the implication of neutrophils in IPF. For example, bad prognosis in IPF patients is associated with an increased number of neutrophils [107], and CXCL8, an important chemokine produced by neutrophils, is also increased during fibrosis [108]. Recently, neutrophil extracellular traps (NETs) have attracted attention as they are meant to be present in Interstitial Lung Disease (ILD) [109]. The role of neutrophils and other immune cells are not discussed here as we are focused on the processes of myofibroblast differentiation. Nevertheless, we recommended a recent review from Desai et al. in which they describe the role of immune cells in IPF very well [110].

## 4. Signaling Pathways in Fibrosis

It is clear then that fibrosis can be described as an imbalance between matrix deposition and matrix degradation, which in turn produces tissue remodelling. Understanding the main signal transduction pathways that govern the differentiation and maintenance of myofibroblasts in fibrotic diseases is therefore of paramount importance for develop anti-fibrotic therapies [111].

The TGF-β signalling: TGF-β regulates varied functions, such as stem cell pluripotency, development, proliferation, differentiation and immune responses. It is one of the most potent factors for inducing myofibroblast differentiation, and has been shown to be produced by AM and AT2 cells [112,113].

Apart from its role in myofibroblast differentiation, TGF-β participates in the induction of many molecules that participate in ECM remodelling, such as matrix metalloproteinases, integrins [114] and in the production suppression of anti-fibrotic molecules such as hepatocyte growth factor and prostaglandin E2 [115,116]. Furthermore, TGF-β inhibits alveolar epithelial cell growth and repair.

An increase in TGF-β levels is seen in many diseases, as well as in most pulmonary diseases such as asthma [117,118], fibrosis and cancer. For example, TGF-β1 expression has been found in small airway epithelial cells among smokers and patients with COPD [119].

The best characterized intracellular effectors of the TGF-β superfamily proteins are the Smads, a group of eight structurally related proteins with homologues that have been identified in both vertebrates and invertebrates [120]. However, recent evidence suggests that TGF-β-independent mechanisms may also contribute to Smad activation in certain cell types. For example, in epithelial cells, it has been suggested that a viral infection activate Smads in a TGF-β-independent manner [121]. The significance of TGF-β-independent Smad signaling in lung fibrosis is not clear yet.

The Wnt/β-catenin signalling: The Wnt–β-catenin system is an evolutionary conserved signalling pathway that is of particular importance for morphogenesis and cell organization during embryogenesis. The system is usually suppressed in adulthood; however, it can be re-activated by organ injury and regeneration [122]. Wnt–β-catenin signalling has been extensively implicated in the differentiation of mesenchymal stem cells and progression of fibrotic diseases [123,124], specifically in the lungs. Hou et al. showed that M2 macrophages can promote myofibroblast differentiation by activating the Wnt–β-catenin signalling pathway [125]. More recently, Cao et al. showed the critical involvement of this pathway in the regulation of myofibroblast differentiation and its importance in pulmonary fibrosis [126].

Many studies have emphasized a key role for canonical Wnt signalling in fibrogenesis of the heart and kidneys, and several fibrotic disorders of the skin [127,128,129]. In the lungs, there is aberrant pathway activation in IPF [123]. Its inhibition has also been related to the development of pulmonary fibrosis [130].

YAP and TAZ signalling: YAP and TAZ are two major molecules in the hippo pathway and have been studied for their role in stem cell fate, organ size control, cancer and very recently mechanobiology [131,132,133,134]. When dysregulated, this pathway is implicated in many diseases, as Cai et al. recently reviewed [135].

There is also increasing evidence of f this pathway’s important role in myofibroblast activation [111]. For example, TAZ upregulation has been seen in fibroblasts from IPF patients, and this increase was related to differentiation to myofibroblast phenotypes [136].

Furthermore, in respiratory fibrotic processes, a study by Liu et al. found that YAP and TAZ were increased in the fibroblasts of IPF patients, whereas their nuclear localization was largely absent in normal lungs [136]. Furthermore, as Gokey et al., recently wrote, many studies have elucidated the upstream and downstream effects of YAP/TAZ in lung fibrosis [137].

We believed that it was important to summarise these three very complex signalling pathways briefly to understand the therapeutically efforts that are currently made to treat lung fibrosis. In that sense it is important to note two different aspects: these pathways are regulated by a very controlled cytosolic/nuclear shuffle that produces the activation of its transcription activators. Second, TGF-β, Wnt, and YAP and TAZ work in concert, instead of in isolation. The final output of these pathways is, of course, the activation and maintenance of the myofibroblast phenotype [111].

## 5. Therapeutic Targets in Lung Tissue Remodelling and Fibrosis

Pulmonary remodelling is the most intractable feature of respiratory diseases because of damage to the structure of the airways and parenchyma, which leads to impaired airflow and gas exchange. Different drugs have been approved for the treatment of diseases (e.g., asthma, COPD, and IPF) that involve pulmonary remodelling, including. However, these treatments do not usually target the molecules that contribute to remodelling. Therefore, drugs that prevent or reverse these events are of potential interest.

### 5.1. Pirfenidone and Nintedanib

In 2014, the U.S. Food and Drug Administration approved pirfenidone and nintedanib, two drugs that proved to be effective in the delay of IPF [138,139]. For example, nintedanib slowed the progression of lung fibrosis and improved lung function in clinical trials [140].

The principal action of pirfenidone is the alteration of the TGF-β pathway, which decreases Smad3, p38 and Akt phosphorylation [141,142]. This inhibition was shown to suppress fibroblast proliferation and collagen production. In addition, pirfenidone also inhibited the production of pro-inflammatory cytokines (TNFα, IL-1, IL-6) and promoted the production of anti-inflammatory cytokines such as IL-10 [143].

On the other hand, nintedanib is a small-molecule tyrosine kinase inhibitor that targets fibroblast migration and proliferation, thereby inhibiting important downstream signaling pathways such as the vascular endothelial growth factor receptor (VEGFR) and the fibroblast growth factor receptor (FGFR).

Due to different mechanisms of action, both pirfenidone and nintedanib can be used in combination. A clinical trial [141] investigated the safety and tolerability of both drugs as a primary end-point. It enrolled 53 IPF patients under combination therapy and compared them to 51 IPF patients treated with nintedanib alone. The adverse effect frequency was similar in the two groups, except for nausea and vomiting which were greater in the combination group. Moreover, the combination treatment was related to a trend in the reduction of forced vital capacity (FVC) [141]. In another study, 89 IPF patients were treated with nintedanib in addition to a pre-existing pirfenidone treatment, but this study did not bring new signals to the known safety profile of either therapy alone [142]. Although the available data are promising for tolerability and safety, more randomized controlled clinical trials should be conducted in large cohorts.

It is important to note that these drugs do not represent a cure, as their benefits are limited to delaying parenchymal remodelling rather than inhibiting or reversing it. For this reason, the search for new therapeutic targets is still a clinical necessity.

### 5.2. Novel Therapies

The activation of multiple pathways that lead to myofibroblast proliferation, migration and differentiation have been identified as possible molecular targets for early clinical trials (Table 2).

#### 5.2.1. Inhibition of Pro-Fibrotic Molecules

In a recent review, Györfi et al. extensively revised the possible strategies for targeting the pro-fibrotic cytokine TGF-β, which, as highlighted above, plays a main role in lung tissue remodeling in pulmonary fibrosis [144]. Apart from pirfenidone, other compounds have been used to suppress parenchyma remodeling by targeting TGF-β. A Phase I, open-label, non-randomized, multicenter study (ClinicalTrials.gov Identifier: NCT00125385) was performed to investigate the tolerability, safety and pharmacokinetics of Fresolimumab (GC1008), a recombinant human monoclonal antibody that inhibits all three TGF-β isoforms [143,144]. However, as far as we are aware, results of this study have never been published.

Many previous studies have described the important role that interleukins play in the pathogenesis of lung fibrosis. IL-13 and IL-14 are key cytokines in many signaling pathways related to lung remodeling [145]. Several monoclonal antibodies targeting IL-13 (tralokinumab) or IL-13 and IL-14 (lebrikizumab) have failed to show efficacy in treating IPF [146,147]. Recently, a study of an IL-11 neutralizing antibody was conducted, and although results were promising, it was only used in vitro or in IPF mice models [148].

Connective tissue growth factor (CTGF, also known as CCN2) is involved in lung remodeling and also mediates TGF-β activation. Pamrevlumab (FG-3019) is a monoclonal antibody against CTGF, and successfully attenuated the progression of IPF in a phase II study [149]. A phase III clinical trial is currently underway (ClinicalTrials.gov Identifier: NCT03955146).

#### 5.2.2. Drugs Targeting ECM Proteins Directly

Several recent novel treatments to inhibit lung fibrosis have been developed that target factors such as ECM proteins that are downstream of TGF-β.

For example, integrins play a central role in ECM adhesion and are involved in fibrosis. Partial inhibition of integrin avβ6 blocks development of pulmonary fibrosis without affecting inflammation [150]. Along that line, a phase II trial (ClinicalTrials.gov Identifier: NCT01371305c) studied the safety and tolerability of a humanized monoclonal antibody (BG00011) against this integrin. However, this trial was terminated due to safety concerns.

Another example is Simtuzumab, a monoclonal antibody against Lysyl oxidase-like-2 LOLX2. This enzyme catalyzes the cross-linking of collagen fibers and elastin [151], and elevated levels of LOXL2 are found in IPF [152]. PXS-5120A, a small molecule inhibitor of LOXL2, has been shown to reduce collagen production although this study was conducted on a bleomycin mouse model [153].

#### 5.2.3. Other Targets

Pentraxin (PTX)-2 has been shown to reduce fibrosis by inhibiting differentiation into profibrotic fibrocytes and TGF-β-producing macrophages [154,155]. Low PTX-2 levels have been observed in patients with IPF [156] and a recombinant human PTX-2 analog (PRM-151) has been shown to ameliorate fibrosis in bleomycin- and TGF-β-overexpressing animal models. In humans, a phase II study demonstrated significant improvement in lung function and stability over 24 weeks with an acceptable safety profile compared to a placebo [157] (ClinicalTrials.gov Identifier: NCT02550873). There is at the moment an ongoing phase III clinical trial (ClinicalTrials.gov Identifier: NCT04594707).

Finally, clinical trials have also been conducted with PBI-4050 (ClinicalTrials.gov Identifier: NCT02538536), an analog of a medium-chain fatty acid that shows an affinity for G-protein receptors that target and inhibit multiple pathways including TGF-β [158]. PBI-4050 alone or in combination with nintedanib gave good results, as the stability of the FVC between baseline and week 12 looked encouraging [159]. Further studies with PBI-4050 in combination with nintedanib or alone are being considered.

Fibrosis and pulmonary remodeling may be treated with multiple drugs in combination, and further studies are needed to identify and better understand the role and therapeutic potential of key molecules in lung fibrosis.

**Table 2 biomedicines-10-01423-t002:** Current clinical trials in idiopathic pulmonary fibrosis (IPF).

Therapy	Mechanism of Action	Clinical Trial Identifier	Status	Observations	Ref.
Fresolimumab	Antibody to neutralize TGFβ	NCT00125385	Phase I completed		-
Lebrikizumab	Anti-IL-13 and IL-14 antibody	NCT01872689	Phase II completed		[147]
Tralokinumab	Anti-IL-13 antibody	NCT01629667	Phase II terminated (lack of efficacy)	The study was stopped due to lack of efficacy	[146]
		NCT02036580	Phase II completed	Early termination due to lack of efficacy	-
Pamrevlumab (FG-3019)	Anti-CTGF antibody	NCT01890265	Phase II completed	Preliminary report shows overall safety and marginally favorable outcome in some patients	[149]
		NCT03955146	Phase III recruiting		-
BG00011 (STX-100)	Anti-integrin antibody	NCT01371305	Phase II completed	This trial was terminated due to safety concerns	-
Simtuzumab	Anti-LOX antibody	NCT01769196	Phase II terminated	The study recommended early termination due to lack of efficacy	[151]
PRM-151	Recombinant human pentraxin 2 (also known as serum amyloid P)	NCT01254409	Phase I	A preliminary report showed overall safety and a marginally favorable outcome in some patients	[157]
		NCT02550873	Phase II completed	Decreased pulmonary function decline and stability in 6MWD over 24 weeks	[160]
		NCT04594707	Phase III recruiting		
PBI-4050	GPR84 antagonist/GPR40 agonist	NCT02538536	Phase II completed	PBI-4050 alone or in combination with nintedanib or pirfenidone showed no safety concerns	[159]

Abbreviations: FVC: forced vital capacity; IL: interleukin; LOX: lysyl oxidase; PDGF: platelet-derived growth factor; TGF-β1: Transforming growth factor 1; IPF: Idiopathic pulmonary fibrosis; 6MWD: 6-min walk distance.

## 6. Final Remarks

Fibrotic phenomena in some lung disorders can be found in isolation or be the pathophysiological basis of devastating diseases such as IPF. Lung myofibroblast likely originate from multiple sources in lung-resident cells. Their relative contributions vary depending not only on the type of injury, but also on the environmental risk factors (cigarette smoking being the most consistent) and normal biological processes such as aging. Both are also related to the differentiation of fibroblasts into myofibroblasts.

Currently, only two drugs are approved to treat IPF, so knowledge of the molecular pathways involved in these complex processes will allow different targets in the same patient to be attacked, as it is being done for other respiratory diseases, such as pulmonary arterial hypertension and cystic fibrosis.

## Figures and Tables

**Figure 1 biomedicines-10-01423-f001:**
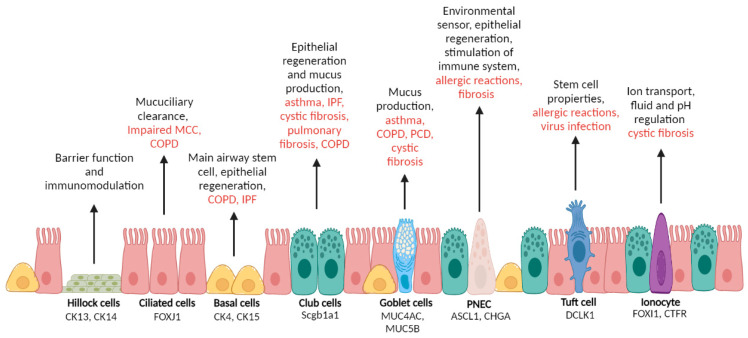
Schematic representation of epithelial cells lining the airways. Cell names are shown in bold together with the main expressed cell markers. The figure also shows the principal known cell role and in red their related respiratory diseases. This figure was created with Biorender (www.biorender.com (accessed on 1 May 2022)).

**Figure 2 biomedicines-10-01423-f002:**
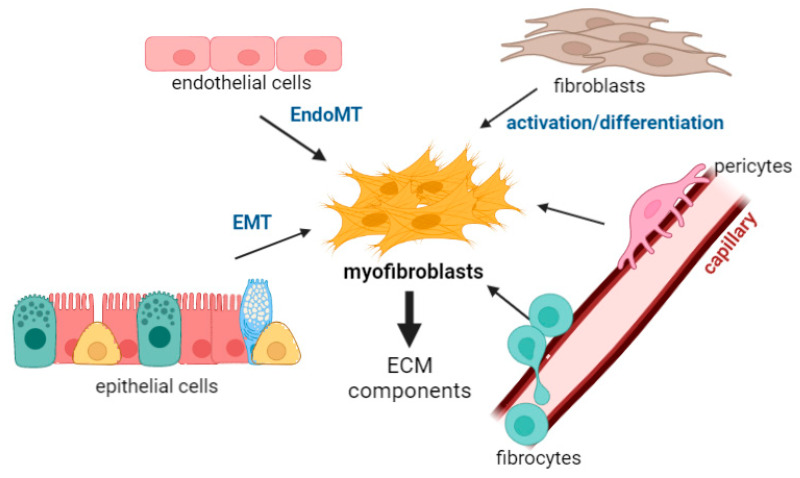
The diagram shows the proposed origin of myofibroblasts: (i) The main myofibroblasts progenitor after injury of different tissues seems to be locally residing fibroblasts, which differentiate into myofibroblasts. (ii) Epithelial and endothelial cells undergo transdifferentiation to form myofibroblasts by epithelial–mesenchymal transition (EMT) or endothelial–mesenchymal transition (EndoMT). (iii) bone marrow-derived precursor cells such as fibrocytes may also differentiate to form myofibroblasts. (iv) pericytes can also differentiate to myofibroblasts. In a normal repair process, excess myofibroblasts undergo regulated apoptosis, but in fibrosis, uncontrolled proliferation of myofibroblasts, along with excess deposition of the ECM, results in structural remodelling that leads to scar lesions, loss of alveolar function, and respiratory failure. This figure was created with BioRender (www.biorender.com (accessed on 1 May 2022)).

## Data Availability

Not applicable.

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
