# Peer review of "Lung Fibrosis and Fibrosis in the Lungs: Is It All about Myofibroblasts?"

_biomedicines, 2022, doi:10.3390/biomedicines10061423_

Round 1

Reviewer 1 Report

Ortiz-Zapater et al. review an interesting topic; lung fibrosis and cellular contributors to it. Here are my suggestions for improving the manuscript:

1-Please provide figure legends for figure 1 and 2.

2-The manuscript may benefit from some style changes; for example, authors don't have to put 9 sections each discussing a type of airway epithelium. Authors also use pronouns extensively; it would be clearer if you mention cell names or abbreviations instead.

3-Table 1. "Macrophages derived secretory proteins": Authors mentioned the proteins but did not mention their roles in fibrosis. Please expand the table to include functions of these molecules in fibrosis. 

4-Authors don't have to remind the reader what they already discussed in the review. For example the following paragraph is redundant: "We have previously discussed how different cells (such as macrophages, fibroblasts, fibrocytes, pericytes and processes of EMT and EndoMT) that populate the respiratory system can both differentiate or activate myofibroblasts."

5-"TGFβ is the most potent factor for the induction of myofibroblast" This and other claims about TGF-beta signaling are debatable. Authors can expand the discussion in light of other reviews. For example PMID: 33160018, PMID: 29355590. 

6-Please correct typos and grammatical mistakes as in "epithelial remodelling see in IPF".

Author Response

Ortiz-Zapater et al. review an interesting topic; lung fibrosis and cellular contributors to it. Here are my suggestions for improving the manuscript:

1-Please provide figure legends for figure 1 and 2.

As the reviewer1 has noted, figure legends were missing and have now been added to the manuscript.

2-The manuscript may benefit from some style changes; for example, authors don't have to put 9 sections each discussing a type of airway epithelium. Authors also use pronouns extensively; it would be clearer if you mention cell names or abbreviations instead.

We thank the reviewer for these comments. We appreciated that maybe there was an excessive detail of the different epithelial cell lines lining the conducting airways, and some changes have made in that section. We have revised the use of pronouns within the manuscript and have removed then in most of the paragraphs as kindly suggested.

3-Table 1. "Macrophages derived secretory proteins": Authors mentioned the proteins but did not mention their roles in fibrosis. Please expand the table to include functions of these molecules in fibrosis. 

We are grateful to the reviewer for this suggestion. We have added another column in the table highlighting the role of the different secretory proteins in fibrosis, providing a more informative and comprehensive table.

4-Authors don't have to remind the reader what they already discussed in the review. For example the following paragraph is redundant: "We have previously discussed how different cells (such as macrophages, fibroblasts, fibrocytes, pericytes and processes of EMT and EndoMT) that populate the respiratory system can both differentiate or activate myofibroblasts."

We added this kind of comment for clarity, but it is also understandable that the paper could benefit from a more impersonal language, so we have removed this kind of “reminders” to the reader within the manuscript.

5-"TGFβ is the most potent factor for the induction of myofibroblast" This and other claims about TGF-beta signaling are debatable. Authors can expand the discussion in light of other reviews. For example PMID: 33160018, PMID: 29355590. 

We thank the reviewer for the review suggestions, that we have read and added to the manuscript. We still believe that TGFb a very important factor for the induction of myofibroblasts. This has been described by many authors before (see for example the papers: Transforming growth factor–β in tissue fibrosis, from Frangogiannis et al, PMID: 32997468, or “The myofibroblast in connective tissue repair and regeneration”, by Hinz et al, in Regenerative Medicine and Biomaterials for the Repair of Connective Tissues, 2010, or even Targeting TGF-β Mediated SMAD Signaling for the Prevention of Fibrosis”, from Walton et al, PMID: 28769795”. We therefore believe that this is an important claim. In any case, we have revised this section carefully as suggested by the reviewer.

  •  

6-Please correct typos and grammatical mistakes as in "epithelial remodelling see in IPF".

We have checked possible typos throughout the manuscript, including the one mentioned above by the reviewer.

Reviewer 2 Report

The authors summarize the available evidence on the pathophysiology of lung fibrosis. The review is well written and the relevant literature is comprehensively covered so that I have only a few minor remarks before the manuscript can be published.

  • I cannot find a heading no. 3.: heading 2 is „Myofibroblasts“, 4 is „Signaling pathways“ – maybe this missing heading should come before „In the next sections, we will describe …“ (= second paragraph on the 4th page).
  • I think that section 3 should be shortened a little as it only describes the physiological role of different cell types. Thereby one would get more space to elaborate more on the pathways in section 4, which is – to my mind – much more interesting.

Author Response

The authors summarize the available evidence on the pathophysiology of lung fibrosis. The review is well written and the relevant literature is comprehensively covered so that I have only a few minor remarks before the manuscript can be published.

  1. I cannot find a heading no. 3.: heading 2 is „Myofibroblasts“, 4 is „Signaling pathways“ – maybe this missing heading should come before „In the next sections, we will describe …“ (= second paragraph on the 4th page).

We are very grateful to the reviewer for having noticed this issue with the heading. This has been now amended.

  1. I think that section 3 should be shortened a little as it only describes the physiological role of different cell types. Thereby one would get more space to elaborate more on the pathways in section 4, which is – to my mind – much more interesting.

As suggested by the reviewer, we have revised the length of the different sections. Sections 2 and 3 have been separated for clarity, and we have elaborated more on the pathways in section 4.

Round 2

Reviewer 1 Report

I would like to thank the authors for their revisions. I have no additional comments.

Author Response

thank you